# PROGRESSIVE COMPRESSED RECORDS: TAKING A BYTE OUT OF DEEP LEARNING DATA

## ABSTRACT

Deep learning training accesses vast amounts of data at high velocity, posing challenges for datasets retrieved over commodity networks and storage devices. We introduce a way to dynamically reduce the overhead of fetching and transporting training data with a method we term *Progressive Compressed Records* (PCRs). PCRs deviate from previous formats by using progressive compression to convert a single dataset into multiple datasets of increasing fidelity—all without adding to the total dataset size. Empirically, we implement PCRs and evaluate them on a wide range of datasets: ImageNet, HAM10000, Stanford Cars, and CelebA-HQ. Our results show that different tasks can tolerate different levels of compression. PCRs use an on-disk layout that enables applications to efficiently and dynamically access appropriate levels of compression at runtime. In turn, we demonstrate that PCRs can seamlessly enable a $2\times$ speedup in training time on average over baseline formats.

## 1 INTRODUCTION

Distributed deep learning exploits parallelism to reduce training time, and consists of three key components: the data pipeline (storage), the forward/backward computation (compute), and the variable synchronization (network). A plethora of work has investigated scaling deep learning from a compute- or network-bound perspective (e.g., Dean et al., 2012; Cui et al., 2016; Abadi et al., 2015; Cui et al., 2014; Jouppi et al., 2017; Lim et al., 2019; Zhu et al., 2018; Alistarh et al., 2017; Lin et al., 2018; Wen et al., 2017; Wangni et al., 2018; Zhang et al., 2017). However, little attention has been paid toward scaling the storage layer, where training starts and training data is sourced.

Unfortunately, hardware trends point to an increasing divide between compute and networking or storage bandwidth (Li et al., 2016; Lim et al., 2019; Kurth et al., 2018). For example, the transportation of data for machine learning is a key factor in the design of modern data centers (Hazelwood et al., 2018), which are expected to be serviced by slow, yet high capacity, storage media for the foreseeable future (David Reinsel, 2018; Cheng et al., 2015; Rosenthal et al., 2012). This, combined with the memory wall—a lack of bandwidth between compute and memory—suggests that, while computation may be sufficient moving forward, the mechanisms for moving data to the compute may not (Wulf & McKee, 1995; Kwon & Rhu, 2018; Hsieh et al., 2017; Zinkevich et al., 2010). The storage pipeline is therefore a natural area to seek improvements in overall training times, which manifest from the storage medium, through the network, and into the compute nodes.

In this work, we propose a novel on-disk format called *Progressive Compressed Records* (PCRs) as a way to reduce the bandwidth cost associated with training over massive datasets. Our approach leverages a compression technique that decomposes each data item into *deltas*, each of which increases data fidelity. PCRs utilize deltas to dynamically compress *entire datasets* at a fidelity suitable for each application's needs, avoiding duplicating the dataset (potentially many times) at various fidelity levels. Applications control the trade-off between dataset size (and, thus, bandwidth) and fidelity, and a careful layout of deltas ensures that data access is efficient at a storage medium level. As a result, we find that for a variety of popular deep learning models and datasets, bandwidth (and therefore training time) can be easily reduced by $2\times$ on average relative to JPEG compression without affecting model accuracy. Overall, we make the following contributions:

1. In experiments with multiple architectures and several large-scale image datasets, we show that neural network training is robust to data compression in terms of test accuracy and training loss; however, the amount of compression that can be tolerated varies across learning tasks.

2. We introduce *Progressive Compressed Records* (PCRs), a novel on-disk format for training data. PCRs combine progressive compression and careful data placement to enable applications to *dynamically* choose the fidelity of the dataset they consume, reducing data bandwidth.

3. We demonstrate that by using PCRs, training speed can be improved by $2\times$ on average over standard formats using JPEG compression. This is achieved by selecting a lower data fidelity, which, in turn, reduces the amount of data read without significantly impairing model performance.

## 2 BACKGROUND

Two complementary concepts make up the process of storing training data: the layout of the data on the storage medium and the representation of the data. Data layout is important because it can help fully utilize the bandwidth potential of the underlying storage system. Data representation is important because it can reduce the amount of data transferred per data unit (i.e., a bandwidth requirement reduction). An example of data representation within the scope of this work is *compression*, which increases the computation per bit—a key property to consider as computation increases faster than bandwidth to storage. Compression may lower image *quality* by introducing artifacts or blur.

**Record Layouts.** Learning from data requires sampling points from a training set, which can cause small, random accesses that are detrimental to the performance of the storage device. Record layouts, such as TensorFlow's TFRecords (TFRecords) or MXNet's ImageRecord (ImageRecord), attempt to alleviate this problem by batching data points together to increase access locality. Batches of training data (i.e., dataset subsets) are then accessed together, amortizing delays in access time across multiple data points. These batches of data are called *records*. The key to any record layout is the *serialization*, which is the conversion of data structures into byte streams. Record designs have different performance properties (e.g., space or access time) when written to disk, as shown in Figure 1.

**Image Compression.** Compressed forms are commonly used to represent training data. JPEG (Wallace, 1992) is one of the most popular formats for image compression and is used ubiquitously in machine learning (e.g., Deng et al., 2009; Russakovsky et al., 2015; Lin et al., 2014; Everingham et al., 2010). Most compression formats (including JPEG) only allow for the compression level, i.e.,

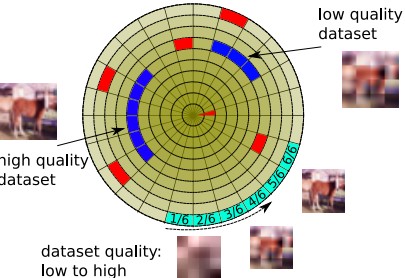

Figure 1: A hard disk with multiple data formats. Other storage media have the same space, bandwidth, and locality considerations. File-per-image formats have highly random behavior. Record formats encode many records at various data qualities to save bandwidth and have sequential behavior for a given fidelity. PCRs maintain the sequential behavior of record formats at multiple fidelities without space overheads.

the trade-off between data size and fidelity, to be set at encoding time, which often results in choosing this level independent of the application. This can result in over-compression, which may negatively impact application convergence quality, or under-compression, which results in excess data size, and thus, slower storage system performance. Worse, deep learning pipelines often involve an application-defined post-processing step (e.g., data augmentation), which may further distort an image and obscure the relationship between image fidelity and model accuracy (Bishop, 1995; Karras et al., 2018; Dziugaite et al., 2016; Arnab et al., 2018). While setting encoding-time parameters is unavoidable, the ability to decompress data as it becomes available (i.e., dynamic compression) provides a means to avoid some of the bandwidth expenses of under-compression by simply terminating decompression once sufficient fidelity is reached.

In Figure 2, we provide a high-level illustration of the JPEG algorithm, which can be customized to support dynamic compression. First, an image is split into blocks of size $8 \times 8$. Each block is converted into the frequency domain, such that frequency 0 is the average color of the block, and higher frequencies encode rapid changes in the block. The low frequencies, such as the average value of the block, store the bulk of the perceptually-relevant content in the image (e.g., knowing the block is mostly blue is more important than knowing a white wave is rippling through it). Quantization,

which discards information from the block and results in compression, thus prioritizes discarding higher frequencies. The resulting quantized table is then serialized into a flat form. Since data is rendered on a screen from left to right, top to bottom, it makes sense to encode the data in this manner, which results in a *sequential* format[1]. Decoding the resulting data is simply a matter of inverting (albeit losslessly) the process that we just described.

**Progressive Image Compression.** *Progressive* formats allow data to be read at varying degrees of compression without duplication. With the sequential case, data is ordered by blocks, and thus, partially reading the data results in "holes" in the image for unread blocks (Wallace, 1992). Dynamic compression ensures that all blocks get some information (*deltas*) before revising them (with more deltas). As progressive formats are simply a different traversal of the quantization matrix, with all else being equal, they contain the same information as sequential JPEG (JPEGTran LibJPEG). Progressive JPEG, combined with an additional rearrangement of data, forms the basis of the idea behind PCRs. In Figure 2, non-progressive formats serialize the image matrix in one pass, while progressive formats serialize the matrix in disjoint groups of deltas which are called *scans*. Scans are ordered by importance (e.g., the first few scans improve fidelity more than subsequent scans). Thus, any references to images generated from scan $n$ will implicitly assume that the image decoder had access to all prior scans (i.e., $\{$scan 1, scan 2, . . . , scan $(n-1)\}$). The bottom of Figure 2 shows how image fidelity improves from a single scan to utilizing all scans.

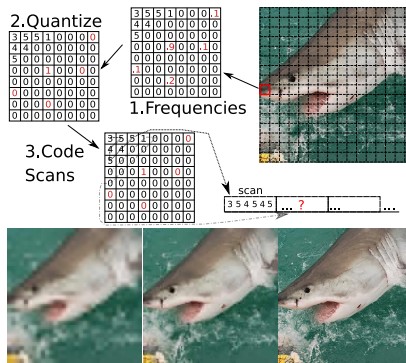

(a) Scan 1    (b) Scan 3    (c) Scan 10

Figure 2: **Top:** JPEG carves an image into blocks, which are then converted into frequencies, quantized, and serialized. Progressive compression writes out a subset of important coefficients from each block before re-visiting the block. **Bottom:** Higher scans have diminishing fidelity returns.

## 3    PROGRESSIVE COMPRESSED RECORDS

In this section, we introduce a novel record format for machine learning training called *Progressive Compressed Records* (PCRs). PCRs are a combination of both layout and data representation. Efficient layouts guarantees that hardware is fully utilized (in terms of bandwidth), while efficient data representations can reduce the total amount of work that is required of the system. To this end, we introduce the concept of *scan groups* in Section 3.1, which leverage both layout and progressive compression to obtain dynamic compression, allowing both high performance reads while reducing the amount of data read. Using progressive compression, scan groups break images into deltas, which are then rearranged in order to facilitate reduced, yet sequential, data access. In Section 3.2, we discuss how PCRs are implemented, covering both creating PCRs (encoding) and reading them (decoding). The benefits of the PCR implementation boiling down to a bit shuffle are that: 1) PCRs are easy to implement, 2) they are fundamentally lossless, and 3) processing them is fast. As we demonstrate in Section 4, while PCRs can be implemented easily, they manifest in large speedups for a variety of scenarios. Further, PCRs can be generalized beyond images and JPEG.

### 3.1    SCAN GROUPS

Scan groups are a collection of scans (deltas) of the same fidelity. Scan groups combine layout with progressive compression to allow reading subsets of the compressed data with high hardware efficiency. PCRs make the assumption that the entire training data will be read at the same fidelity. Using this assumption, scan groups rearrange the data such that all deltas of the same fidelity are grouped together. This, in turn, enables groups of deltas to be read together sequentially, which creates dynamicity in the decoding process. Since scans are sorted by importance, and scan groups are a set of scans, the scan groups are also sorted by importance.

To paint a clear representation of how scan groups work, we point the reader to Figure 3. PCRs begin with some metadata which is assumed to be needed by all machine learning tasks, such as labels or

---

[1]"Sequential" refers to in-memory and should not be confused with sequential on-disk access.

bounding boxes. In practice, metadata is small in size, and, thus, the space overheads are negligible. The metadata is followed by scan groups, which consist of scans. The scan 1 representation of the shark in Figure 2 will be available in its record once data is read up to offset 1. Likewise, the scan 3 representation will be available once the record is read up to offset 3, and the representation will be more crisp as 3 scans were used per image, rather than 1. Reading up to the end of the record yields the most complete representation of the image. As scan groups consist of groups of the same fidelity, every image contained in a record is available at the same fidelity at the same group offset. Users of PCRs can read data at a certain scan fidelity by simply reading the on-disk byte stream from the start of the PCR (i.e., offset 0) to the byte offset corresponding to the corresponding scan group. Partially reading the records results in bandwidth savings without re-encoding the data.

## 3.2 IMPLEMENTATION

There are two fundamental PCR implementation details: the encoding process and the decoding process. The encoding process transforms a set of JPEG files into a directory, which contains 1) a database for PCR metadata and 2) at least one `.pcr` file. The decoding process, which takes the directory as input and yields a set of JPEG images, efficiently inverts a subset of the encoding. The dataset is split into many PCRs, and, thus, the training process is reading tens to hundreds of `.pcr` files per epoch. The data loader is where the PCR decoding library interfaces with the inputs provided to deep learning libraries (e.g., TensorFlow (Abadi et al., 2015), MXNet (Chen et al., 2015), PyTorch (Paszke et al., 2017)). Below, we describe how each of these steps is done.

**Encoding.** Given a set of images, the PCR encoder must break the images into scans, group the scans into scan groups, and sort the scan groups by fidelity. Once the groups are sorted, the PCR encoder can serialize the groups while taking note of their offsets (so that subsets may later be decoded). The metadata (e.g., labels) is prepended to the serialized representation, and the serialized representation is written to disk. We focus on grouping

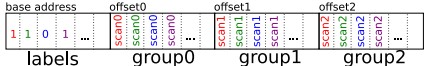

Figure 3: PCRs encode label metadata followed by all scan groups. Accessing the dataset at a lower fidelity requires reading up to a certain address offset.

JPEG due to its generality, but PCRs can use any dataset-level progressive format. Images can be decomposed in both space and fidelity; other data modalities (e.g., video) may also have time.

Our implementation uses JPEGTRAN (JPEGTran Man Page) to losslessly transform the set of JPEG images into a set of progressive JPEG images. With the default settings, each JPEG is broken up into 10 scans. The encoder scans the binary representation of the progressive JPEG files, searching for the markers that designate the end of a scan group. The encoder thus has access to all 10 offsets within the JPEG files that can be used to determine the boundaries between scan regions. Forming scan groups requires grouping the scan regions with the same fidelity together, which can be done in one pass over the set of images corresponding to that PCR. This grouping must be reversible, as the decoding process will un-group the scans to reconstruct the original images. This grouping can be done with existing serialization libraries. We use Protobuf (Protobuf) to serialize the groups as well as the labels. However, it is key that every group (and the metadata) be serialized as a separate message, as Protobuf can rearrange the contents within a message, and thus can rearrange the ordering of the groups themselves. We finally concatenate the contents of the messages and write them out as one file. As shown in Appendix A.5, any record format conversion can be expensive; PCRs benefit from requiring only a single conversion for multiple tasks.

**Decoding.** To decode a PCR file, one has to first lookup the file's scan group offsets in the database. The offsets provide sufficient information to do a partial read of the file (e.g., instead of reading the entire file, we read only enough bytes to read up to the desired scan group). Decoding the JPEGs requires inverting the PCR scan-group grouping process for the available scan-groups prior to JPEG decode. Since we are missing scan-groups, we terminate the byte stream with an End-of-Image (EOI) JPEG token—this technique allows most JPEG decoders to render the byte stream with only the available subset of scans. The bulk of the inverse conversion is done in 150 lines of C++ code.

**Loader.** We implemented PCR loaders using PyTorch's dataloader as well as DALI (NVIDIA, 2018)'s ExternalSource operator to return batches of images at a configurable fidelity (with the corresponding labels). We find that a *pipeline* abstraction simplifies loader design, since record-based datasets can be easily iterated sequentially. In contrast, the PyTorch Dataloader abstrac-

tion, which assumes that we can index randomly into an in-memory data structure (e.g., `i = RandInt(0, n); (x, y) = data[i];`), is harder to use for constantly fetching record formats off disk. Our implementation, while being only several hundred lines of code, obtains image rates that are competitive (e.g., faster/slower depending on number of scans) with the included DALI TFRecord loader, showing that PCRs can be implemented efficiently (i.e., fast enough to rarely bottleneck data loading) with a low amount of engineering effort.

## 4 EXPERIMENTS

This section presents our evaluation of PCRs using a suite of large-scale image datasets. As large images are more taxing to a system's network and storage, our evaluation focuses on datasets with high-resolution images. We describe our experimental setup in Section 4.1. We present our evaluation results in Section 4.2, showing that halving data bandwidth per image results in comparable accuracy but with half the training time. In Section 4.3, we analyze the intuitive relationship between objective measures of image fidelity and time-to-accuracy. Finally, in Section 4.4, we present results that trace the training time speedups to the data loading times themselves.

### 4.1 EVALUATION SETUP

Our evaluation uses the ImageNet ILSVRC (Deng et al., 2009; Russakovsky et al., 2015), HAM10000 (Tschandl et al., 2018), Stanford Cars (Krause et al., 2013), and CelebA-HQ (Karras et al., 2018) datasets, which are described below. See Appendix A.4 for additional details.

**Datasets.**

- **ImageNet-100** ImageNet provides a wide diversity of classes, of which we focus on the first 100 to make training times more tractable. Since classes are roughly ordered by ImageNet categories, this results in a fine-grained, i.e., hard to classify, multiclass task. We convert the dataset into PCRs in batches of 1024, which results in 126 PCRs. We use the full ImageNet dataset in Appendix A.7.

- **HAM10000** We split the HAM10000 dataset randomly 80%/20% between train and test. We convert the dataset into PCRs in batches of 64, which results in 125 PCRs of similar size as the ones used for ImageNet-100.

- **Stanford Cars** The Stanford Cars dataset is another fine-grained classification dataset, since all images are cars, and there are 196 classes spread over 16k images. We believe this dataset highlights some of the worst-case training scenarios, as it is considerably easier to predict highly compressed variants of unrelated images by exploiting low frequency image statistics (e.g., planes vs. frogs). We explore a coarse-grained version of Cars in Appendix A.6. Cars has 63 PCRs.

- **CelebAHQ-Smile** CelebA-HQ is a high-resolution derivative of the CelebA dataset (Liu et al., 2015), which consists of 30k celebrity faces at $1024^2$. We use the annotations provided by CelebA to construct a smiling or not smiling dataset. We split the 30k dataset into 80%/20% train/test, and we convert the training set into 93 PCRs.

All datasets utilize resizing, crop, and horizontal-flip augmentations, as is standard for ImageNet training. We provide examples of scan groups for these datasets in Appendix A.8.

**Training Regime.** We use pretrained ImageNet weights for HAM10000 and Cars due to the limited amount of training data. We use standard ImageNet training, starting the learning rate at $0.1$ (with gradual warmup (Goyal et al., 2017)) and dropping it on epoch 30 and 60 by $10\times$. After augmentations, all inputs are of size $224 \times 224$. The pretrained experiments (HAM10000 and Cars) start at a learning rate of $0.01$ to avoid changing the initialization too aggressively. We use `fp16` training (Micikevicius et al., 2018) as it results in an additional $10\%$ images per second (see Appendix A.3). We use a ResNet18 (He et al., 2016) and ShuffleNetv2 (Ma et al., 2018) architecture for our experiments with a batch size of 128 per each worker. We run each experiment at least 3 times to obtain confidence intervals given different random seeds and sources of non-determinism such as multi-threading and I/O.

**System Setup.** We run distributed experiments on a 16-node Ceph (Weil et al., 2006) cluster connected with a Cisco Nexus 3264-Q 64-port QSFP+ 40GbE switch. Each node has a 16–core Intel E5–2698Bv3 Xeon 2GHz CPU, 64GiB RAM, NVIDIA TitanX, 4TB 7200RPM Seagate

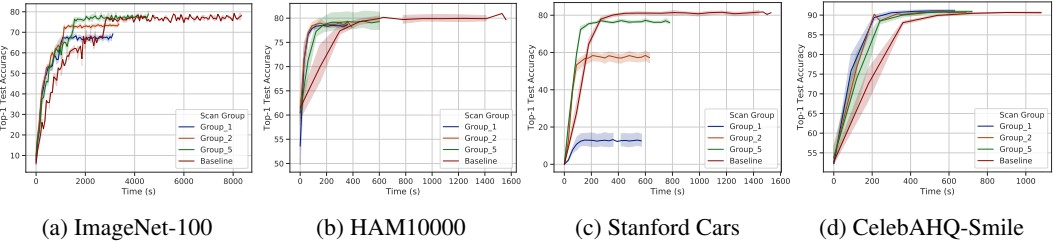

(a) ImageNet-100      (b) HAM10000      (c) Stanford Cars      (d) CelebAHQ-Smile

Figure 4: Top-1 test performance with ResNet-18. Time is the x-axis (seconds) and is relative to first epoch. 95% confidence intervals are shown. Higher scan groups are less compressed.

ST4000NM0023 HDD, and a Mellanox MCX314A-BCCT 40GbE NIC. All nodes run Linux kernel 4.15 on Ubuntu 18.04, CUDA10, and the Luminous release (v12.2.12) of Ceph. We use six of the nodes as Ceph nodes; five nodes are dedicated as storage nodes in the form of Object Storage Devices (OSDs), and one node is used as a Ceph metadata server (MDS). The remaining 10 nodes are used as machine learning workers for the training process. This means there is a 2:1 ratio between compute and storage nodes. We use PyTorch (Paszke et al., 2017) (v1.12) with NVIDIA Apex (Apex) (v0.1) and NVIDIA DALI (NVIDIA, 2018) (v0.14.0). We use at least four worker threads to prefetch data in the loader. While we focus on this particular distributed setting, we observe similar time-to-accuracy gains on a single machine with eight GPUs sharing the same disk, and we believe the results will generalize to different setups.

## 4.2 TIME TO ACCURACY

The time-to-accuracy results for ResNet18 training are presented in Figure 4, while those of ShuffleNetv2 are presented in Figure 6. See Appendix A.2 for a tabular view and Appendix A.1 for the corresponding training loss results. All scan groups within a dataset were run for the same amount of epochs, so lower scan groups finish earlier. 90 epochs are shown for ImageNet, 150 epochs are shown for HAM10000, 250 epochs are shown for Stanford Cars, and 90 epochs are shown for CelebAHQ-Smile. We sample the test accuracy every 15 epochs for non-ImageNet datasets to reduce interference with training measurements. To avoid measuring implementation differences with other loaders, our evaluation focuses on the differences obtained by reading various amounts of scan groups. Reading all the data (up to scan group 10) is the baseline.

First, we note that for all datasets, except for Cars, PCRs provide a $2\times$ boost to time-to-accuracy compared to the baseline. The reason for this speedup is that lower scan groups are smaller. As shown in Figure 5, scan group 5 is roughly half the size of the baseline, and scan group 1 is a fifth of scan group 5 (i.e., a potential $10\times$ bandwidth savings). This trend holds across datasets (see Appendix A.1). As we will discuss in Section 4.4, the space savings manifest in reduced dataloader latencies. Second, we note that there is an inherent trade-off between convergence quality and the speedup attained by using less storage resources. In general, although lower fidelity scan groups allow the system to operate more efficiently, they do so at the expense of model convergence. Scan group 1, the lowest fidelity scan, performs poorly, especially on Cars,

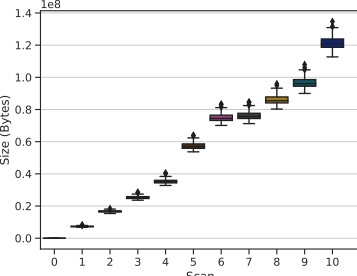

Figure 5: The size in bytes of scan groups for ImageNet-100.

where fine-grained details are important. Scan groups limit the maximum achievable accuracy on a task; if learning plateaus prematurely, applications should raise the scan group in a manner similar to dropping the learning rate. Third, the relative rankings of scan groups are relatively stable across models and datasets, which reduces tuning efforts in choosing the appropriate scan group. We further relate these rankings to the fidelity of the scan groups in Section 4.3. Our conclusion is that, for most datasets, scan group 5 costs half as much in terms of bandwidth, but reaches the same level of test accuracy as the baseline—thus, it is a good default. This is most apparent for ImageNet and HAM10000, which are challenging enough for small variations in image fidelity to make a commensurate difference in test accuracy. In contrast, Cars is too fine-grained to allow images to be degraded, and CelebAHQ-Smile is too coarse-grained for image degradation to matter.

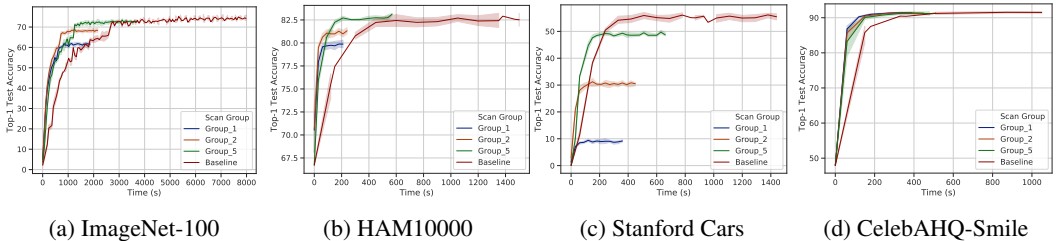

Figure 6: Top-1 test performance with ShuffleNetv2. Time is the x-axis (seconds) and is relative to first epoch. 95% confidence intervals are shown. Higher scan groups are less compressed.

## 4.3 THE RELATIONSHIP BETWEEN IMAGE FIDELITY AND TEST ACCURACY

We use MSSIM (Wang et al., 2003), a standard measure of image similarity, to compare how various scans approximate the reference image, and we show the results in Figure 7. We find that there is a strong connection between MSSIM and the resulting final test accuracy, especially when comparing scan groups *within* a task. Our preliminary tests demonstrate that scan groups that have very similar MSSIM perform very similarly, which is why only groups 1, 2, 5, and the baseline are shown. Due to the way progressive JPEG is coded by default, groups tend to cluster (e.g., 2, 3, and 4 are usually similar, while 5 introduces a difference). We note that MSSIM being poor (e.g., scan group 1 for cars) or MSSIM being close to baseline (e.g., scan group 5 for HAM10000) are good predictors of relative test accuracy within tasks. MSSIM can therefore be used as a diagnostic for choosing scans.

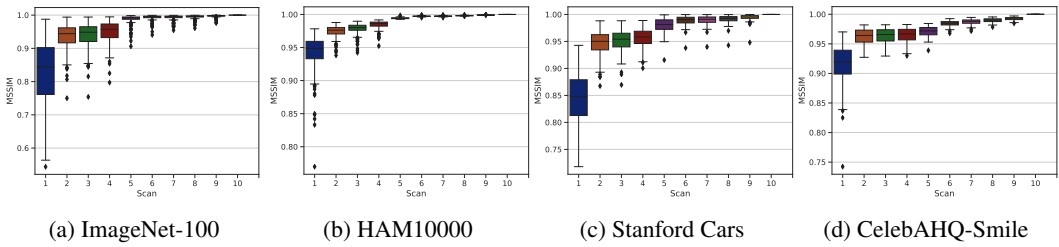

Figure 7: The reconstruction quality (measured with MSSIM) of using various amounts of scans. Perfect reconstruction has an MSSIM of 1. Higher scans have diminishing fidelity returns.

## 4.4 THE RELATIONSHIP BETWEEN SCANS AND DATA STALLS

The datasets we evaluated show that data loading can slow down the training process. To highlight these slowdowns, and the improvements PCRs achieve by not using all scan groups, we present the loading time of data for the ResNet18 ImageNet-100 run in Figure 8. We obtain similar results for the other datasets. The baseline of using all scan group results in high periodic loading stalls, where the prefetching queue was drained. Upon blocking, training cannot proceed until the worker threads obtain a full batch of data. Periods of (mostly) no stalls are caused by both threads pre-fetching the data and single records servicing multiple minibatches. Using fewer scan groups reduces the amount of data read, which results in lower magnitude stalls. We observe these stalls with both DALI and PyTorch loaders.

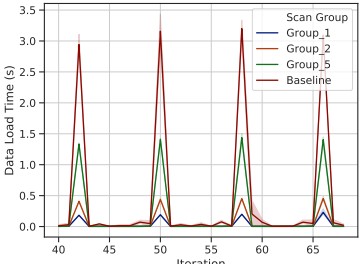

Figure 8: Data loading stalls are periodic and are followed by extents of prefetched data. Lower scan groups reduce stall time.

## 5 RELATED WORK

**Training Over Large Datasets.** Training with massive amounts of parallelism (thus stressing system bandwidth) while achieving near-linear speedup has been the focus of previous work, and it highlights a practical need for efficient data pipelines at scale. A common objective is training mod-

els over ImageNet in a record amount of time (Goyal et al., 2017; You et al., 2018; Jia et al., 2018; Ying et al., 2018; Yamazaki et al., 2019). This line of work, while demonstrating immense bandwidth needs, typically keeps data in memory, avoiding storage problems altogether. Recently, the high performance computing community has become interested in training models at massive scale (27k GPUs) (Kurth et al., 2018). Since each GPU matches a disk in bandwidth, the dataset was partitioned among the local memory/storage of the nodes, avoiding the distributed filesystem. Our work attempts to reduce the storage bottleneck altogether, such that anything from a couple disks to a distributed file system could service many GPUs. A separate line of work shows that I/O is a significant bottleneck for certain tasks and proposes optimizing I/O via a set of deep-learning specific optimization to LMDB (Pumma et al., 2019). In contrast, our focus is more on data representation, which is independent of the internals of the storage system. Production systems such as TFX (Baylor et al., 2017) have used custom Protobuf parsers to get 2–5× speedups for simple (e.g., linear) models; these techniques are complementary to ours and reduce loader computational overheads.

**Dataset Reduction Techniques.** The availability of larger datasets has spawned interest in learning algorithms that guaranteed both "good" model accuracy and lower computational complexity. Data reduction techniques, such as sketching, coresets, clustering, and sampling, have been used to reduce the size of a training set (Karnin & Liberty, 2019; Feldman et al., 2013; Liberty, 2013; Woodruff, 2014; Daniely et al., 2017; Kabkab et al., 2016; Bachem et al., 2017). A different approach is to use the unaltered training set, but reduce the size of the *active* training set to reduce bandwidth requirements (Matsushima et al., 2012). In contrast, we modify the data representation and layout to be more efficient across a wide variety of models.

**Compression.** Finally, the reduction of data size via compression methods is ubiquitous across computer systems. To avoid costly model transmission/storage, prior work compressed neural network models (Han et al., 2016b;a; 2015; Cheng et al., 2017; Xu et al., 2018; Hwang & Sung, 2014; Anwar et al., 2015; Denton et al., 2014). Similarly, dataset distillation (Wang et al., 2018) compresses a model's parameters into a few training examples. Our work attempts to compress data for training, and not the network itself. Prior work has looked into optimizing training systems by compressing neural network training network traffic (Lim et al., 2019; Alistarh et al., 2017; Lin et al., 2018; Wen et al., 2017; Wangni et al., 2018; Zhang et al., 2017). This trend is not specific to machine learning; prior work in databases, computer memories, and the web used compression to reduce system bandwidth requirements (Zukowski et al., 2006; Abadi et al., 2006; Pekhimenko et al., 2018; 2012; Yan et al., 2017; Agababov et al., 2015). Our work focuses on bandwidth for ML data pipelines by utilizing the compression robustness found in most models. Other work modifies models to be able to directly train on compressed representations for the purpose of avoiding decoding or reducing model complexity (Gueguen et al., 2018; Torfason et al., 2018; Fu & Guimaraes, 2016; Ulicny & Dahyot, 2017). Our work differs in motivation, as we do not focus on model computation or make modifications to the models. Previous work has investigated how image degradation (e.g., JPEG artifacts) affect inference (Dodge & Karam, 2016; Vasiljevic et al., 2016; Peng et al., 2016; Zheng et al., 2016); in contrast, our work focuses on the effects of compression on training.

## 6    CONCLUSION

To continue making advances in machine learning, researchers will need access to larger and larger datasets, which will eventually spill into (potentially distributed) storage systems. Storage and networking bandwidth, which are precious resources, can be better utilized with efficient compression formats. We introduce a novel record format, *Progressive Compressed Records* (PCRs), that trades off data fidelity with storage and network demands, allowing the same model to be trained with 2× less storage bandwidth while retaining model accuracy. PCRs use progressive compression to split training examples into multiple examples of increasingly higher fidelity without the overheads of naive approaches. PCRs avoid duplicating space, are easy to implement, and can be applied to a broad range of tasks dynamically. While we apply our format in this work specifically to images with JPEG compression, PCRs are general enough to handle various data modalities or additional compression techniques; future work will include exploring these directions in fields outside of visual classification, such as audio generation or video segmentation.

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

# A APPENDIX

## A.1 LOSS, SPACE SAVINGS, AND ACCURACY PER EPOCH

Below, we provide additional experiment plots that were omitted in the main text. Figure 9 and Figure 10 contain the loss over time for the ResNet-18 and ShuffleNetv2 experiments shown in Section 4. Figure 11 extends Figure 5 to show the scan sizes for all datasets. It is worth noting that Top-5 accuracies mirror the Top-1 accuracies trends for ImageNet and Cars.

To measure the effect of compression without accounting for time, we show accuracy vs. epoch plots in Figure 12 and Figure 13. While compression can itself be viewed as a data augmentation (e.g., removing high frequency features that can possibly cause overfitting), we notice that it does not usually improve accuracy. Rather, most of the gains in time-to-accuracy are from faster image rates.

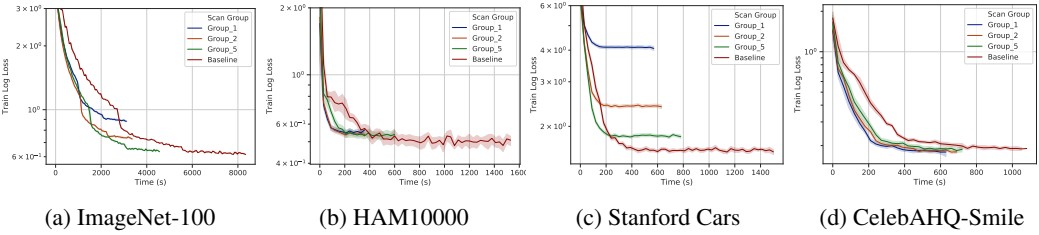

|  (a) ImageNet-100 | (b) HAM10000 | (c) Stanford Cars | (d) CelebAHQ-Smile |

Figure 9: Training loss with ResNet-18. Time is the x-axis (seconds) and is relative to first epoch. 95% confidence intervals are shown.

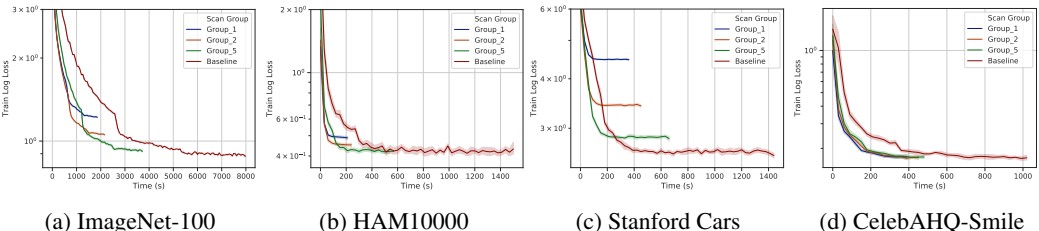

|  (a) ImageNet-100 | (b) HAM10000 | (c) Stanford Cars | (d) CelebAHQ-Smile |

Figure 10: Training loss with ShuffleNetv2. Time is the x-axis (seconds) and is relative to first epoch. 95% confidence intervals are shown.

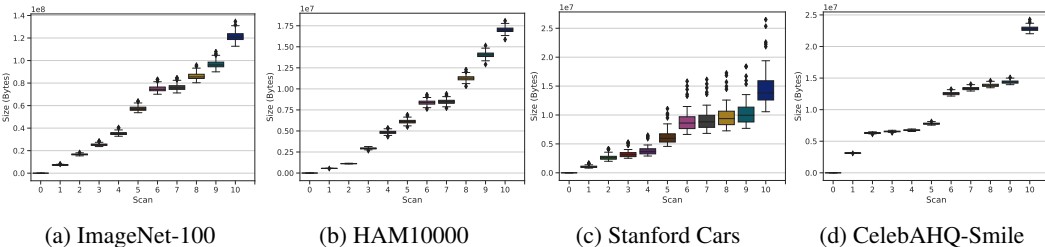

|  (a) ImageNet-100 | (b) HAM10000 | (c) Stanford Cars | (d) CelebAHQ-Smile |

Figure 11: The size in bytes of various levels of scans read. Scan group 0 is shown, which contains only labels and is typically ∼100 bytes. Each scan adds roughly a constant amount of data (i.e., linear scaling), although certain scans add considerably more than others (i.e., sizes sometimes cluster) due to techniques like chroma subsampling. Using all 10 scans can require over an order of magnitude more bandwidth than 1–2 scans.

## A.2 TIME TO CONVERGENCE TABLE

We provide a table of time-to-accuracy in Table 1 to help with reading Figure 4 and Figure 6. For Stanford Cars, low numbers of scans do reach accuracies faster than the baseline, but there is a

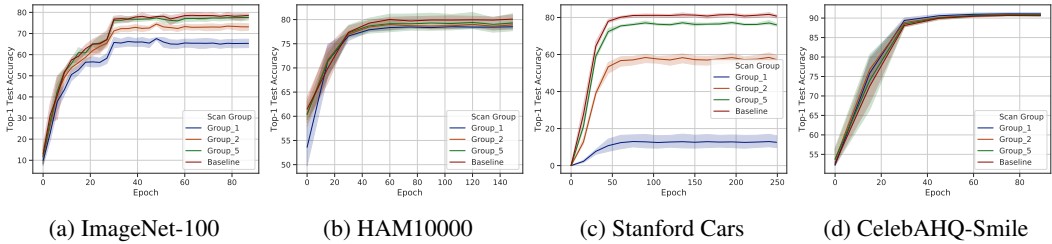

| (a) ImageNet-100 | (b) HAM10000 | (c) Stanford Cars | (d) CelebAHQ-Smile |

Figure 12: Testing accuracy with ResNet-18. Epochs are the x-axis. 95% confidence intervals are shown.

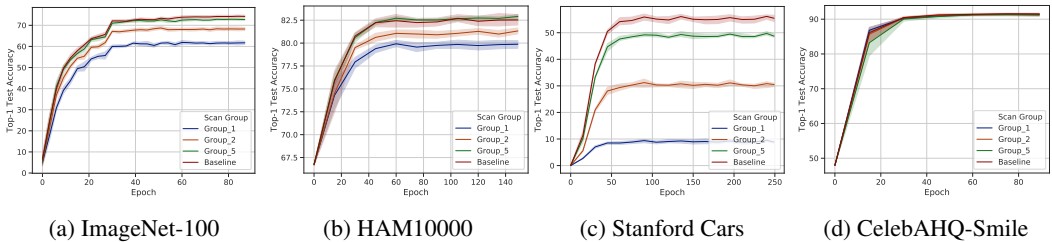

| (a) ImageNet-100 | (b) HAM10000 | (c) Stanford Cars | (d) CelebAHQ-Smile |

Figure 13: Testing accuracy with ShuffleNetv2. Epochs are the x-axis. 95% confidence intervals are shown.

noticeable drop in accuracy. This issue of achieving comparable accuracy for the Cars dataset is further explored in Appendix A.6.

| Dataset | Model | Accuracy Goal | Baseline Time (s) | PCR Time (s) |
|---|---|---|---|---|
| **ImageNet-100** | ResNet-18 | 76.2% | 2660 | 1500 |
| **HAM10000** | ResNet-18 | 78.9% | 420 | 142 |
| **Stanford Cars** | ResNet-18 | 81.5% | 810 | N/A |
| **CelebAHQ-Smile** | ResNet-18 | 90.6% | 825 | 305 |
| **ImageNet-100** | ShuffleNetv2 | 72% | 2690 | 1500 |
| **HAM10000** | ShuffleNetv2 | 82.7% | 1050 | 210 |
| **Stanford Cars** | ShuffleNetv2 | 55.9% | 510 | N/A |
| **CelebAHQ-Smile** | ShuffleNetv2 | 91.4% | 710 | 325 |

Table 1: Progressive compression time to accuracy for ResNet-18 (top) and ShuffleNetv2 (bottom) compared to the baseline. Using less scans can reduce training times while still hitting the goal accuracy.

## A.3 EXPERIMENT SETUP

Below we describe details of how the experiments were run, such as hardware characteristics and software configurations.

**Benchmark Cluster Speeds.** As noted in the main text, we utilize a NVIDIA TitanX Graphics Processing Unit (GPU) on each node for the model training. This GPU allows us to train (with `FP32`/`FP16`) ResNet-18 at 405/445 images per second and ShuffleNetv2 at 760/750 images per second. With a cached, decoded dataset of $224 \times 224$ resolution images, we achieve a cluster-wide 3625/4050 images per second for ResNet-18 and 6975/7075 images per second for ShuffleNetv2. ImageNet images are around 110kB on average; with 10 GPUs, the cluster can consume 445 megabytes/s (ResNet-18) and 775 megabytes/s (ShuffleNetv2) of storage system bandwidth. GPUs continue to get faster over time, and faster GPUs (or other accelerators) have higher I/O bandwidth demands.

**Decoding Overhead.** Progressive compression has some computational overhead associated with decompression compared to baseline formats. This overhead can grow with the number of scans, and, thus, users of PCRs may be concerned about the trade-offs between decoding overheads and bandwidth savings. First, we note that PCRs can use a large number of scans (e.g., hundreds), but, in practice, useful behavior is observed using only 10 scans (of which we only use 4). Second, the decoding overhead is often a favorable trade-off compared to a storage bottleneck, if one exists. To test this, we a Python microbenchmark that stores a subset of ImageNet data in memory and uses the PIL and OpenCV libraries for decoding. For PIL, we process 230 baseline images per second and 150 progressive images per second. For OpenCV, we process 225 baseline images per second and 165 progressive images per second. Thus, progressive compression with 10 scans adds only around 40–50% computational expense over baseline formats for common implementations. This speed, combined with additional optimizations such as multi-core parallelism (e.g., we would expect $4\times$ these rates with 4 cores), suggests that while decoding can be an issue, the penalty from using progressive images can be managed more easily than a storage bottleneck (i.e., compute can usually be traded for storage bandwidth). Further, some of the decoding can actually be moved to an accelerator, like the GPU used for training, something which is already available via nvJPEG[2]. Reducing this computational expense by optimizing the implementation or reducing the amount of scans (since our experiments only use 4 distinct scans) is left as future work.

**Image Loading Rates.** We provide image loading rates observed during training in Table 2. Using more scans slows down training significantly, as can be seen in the image rates. It is worth noting that these rates vary considerably during runtime (due to stalls), and ShuffleNetv2 is capable of a higher maximum training rate than ResNet-18. Further, as the number of scans is reduced, image rates approach the maximum achievable by the cluster for each model.

| Dataset | Model | Scan 1 | Scan 2 | Scan 5 | Baseline |
|---------|-------|--------|--------|--------|----------|
| **ImageNet-100** | ResNet-18 | 3825 | 3600 | 2800 | 1700 |
| **HAM10000** | ResNet-18 | 3600 | 3500 | 2500 | 925 |
| **Stanford Cars** | ResNet-18 | 3700 | 3500 | 2900 | 2000 |
| **CelebAHQ-Smile** | ResNet-18 | 3500 | 3100 | 2950 | 1925 |
| **ImageNet-100** | ShuffleNetv2 | 6300 | 5650 | 3650 | 1575 |
| **HAM10000** | ShuffleNetv2 | 6000 | 5500 | 2500 | 900 |
| **Stanford Cars** | ShuffleNetv2 | 6100 | 4900 | 4000 | 1800 |
| **CelebAHQ-Smile** | ShuffleNetv2 | 5500 | 4600 | 4300 | 1925 |

Table 2: PCR image rates for ResNet-18 (top) and ShuffleNetv2 (bottom). With data in memory, the 10 worker cluster can support training with 4050 images per second for ResNet-18 and 7075 images per second for ShuffleNetv2. Using less scans approaches this maximum training rate.

## A.4 DATASET DETAILS

Below we describe the characteristics of the used datasets.

**ImageNet-100 Creation.** The ImageNet-100 dataset was constructed by subsampling 100 classes out of the 1000 classes found in the ImageNet ILSVRC dataset (Deng et al., 2009; Russakovsky et al., 2015). These classes were chosen arbitrarily to limit computation time—they are the first 100 classes of ImageNet in sorted directory listing form i.e., `n01440764-n01855672`.

**CelebAHQ-Smile Creation.** The CelebAHQ dataset (Karras et al., 2018) was created as a high quality version of the CelebA dataset (Liu et al., 2015). CelebA contains attributes for each face, such as whether the face is smiling or not. CelebAHQ-Smile utilizes these attributes to construct a dataset of 30k faces, where each face is assigned a binary variable for smiling or not. While the CelebA dataset was subsampled to construct CelebAHQ, we do not subsample CelebAHQ further (i.e., we use all 30k images it contains).

**Record and Image Quality Details.** We provide the dataset size details for the encoded datasets in Table 3. As the original (e.g., lossless) images are hard to find, we estimate the JPEQ qual-

---

[2]`https://developer.nvidia.com/nvjpeg`

ity setting of the training set with ImageMagick using `identify -format '%Q'`. The JPEG quality setting determines the level of frequency quantization outlined in Figure 2. Intuitively, one would expect that higher quality JPEG images could allow more aggressive PCR compression rates for a fixed resolution, since each image has more redundant information on average. ImageNet and HAM10000 both have high quality images. CelebAHQ has lower quality images, but they are downscaled to $256 \times 256$ for training purposes, which increases the information density in the image (e.g., blurry images can be made to appear less blurry by downsampling), a fact exploited in prior work (Yan et al., 2017). Cars is neither high JPEG quality or large resolution. Under-compressing images (perhaps at high resolution) during the initial JPEG compression may allow for a larger range of viable scan groups.

| Dataset | Record Count | Image Count | Dataset Size | Quality |
|---|---|---|---|---|
| **ImageNet-100** | 126 | 126763 | 15GB | 92.8% |
| **HAM10000** | 125 | 8012 | 2GB | 100% |
| **Stanford Cars** | 63 | 8144 | 887MB | 83.8% |
| **CelebAHQ** | 93 | 24000 | 2GB | 75% |
| **ImageNet-Full** | 1251 | 1281167 | 129GB | 91.7% |

Table 3: PCR dataset size and record count information. Datasets vary in terms of number of images and the size of images. Some datasets, such as HAM10000, have image sizes larger than average.

## A.5 RECORD FORMAT CONVERSION TIMES

We provide bandwidth-optimized record baselines in Figure 14, where we re-encode the images using a statically-chosen level of compression. These baselines re-encode the images with 50% quality and 90% JPEG quality, respectively, to reduce dataset size at a *fixed* level of fidelity. It is worth noting that re-encoding images compounds with the original JPEG compression, so the re-encoded image quality may be lower than 50% or 90% quality compared to the images in their original lossless form. This is in contrast to PCRs, which losslessly convert the images into a progressive format, which allows *dynamic* access to the level of fidelity. We observe that both the baseline method of dataset bandwidth reduction and the PCR method can take considerable encoding time, since the encoding time scales proportionally to the dataset size. We also observe that the PCR method is competitive ($1.15\times$ to $2.98\times$) to that of the baseline in terms of encoding time. PCRs avoid having to re-encode a dataset at multiple fidelity levels, and, therefore, they can save both storage space and encoding time. Converting the full ImageNet into record format takes roughly $16\times$ longer than the 6 minutes needed for the $10\times$ smaller subsampled dataset—the PCR conversion is 96 minutes (53 minutes are spent in JPEG conversion). One reason for this additional slowdown is that any system caches (e.g., in the distributed filesystem or the file cache on the converter node) are less likely to see a cache hit due to the working set size being larger. Although the exact conversion times are dependent on implementation, hardware, and the dataset, conversions times can be in the range of one hour of compute time per 100 GB.

## A.6 COARSE GRAINED VS. FINE GRAINED CARS EXPERIMENTS

We provide experiments validating that compression needs vary within the same dataset for different tasks in Figure 15 and Figure 16, which show accuracy and loss, respectively. This experiment simply coarsens the granularity of the classification task, and demonstrates that lower scan groups can be used for tasks which are easier. The full range of classes is used for *Baseline* (i.e, car make, model, and year create a unique class), only car make is used for *Make-Only*, and a binary classification task of Corvette detection is used for *Is-Corvette*. We can see that compared to the original task, the coarse tasks reduce the gap between scan groups, and the binary task closes the gap even more. This suggests that as the tasks get easier, the tolerance for lower scan groups grows. Simply re-assigning the class labels to a coarser class reduces the complexity of the task and closes the accuracy gap across scan groups. A fixed PCR record encoding (i.e., without re-encoding) can support multiple tasks at the optimal quality, whereas static approaches may need one encoding per task. Some training methodologies, such as Progressive GAN training (Karras et al., 2018), utilize

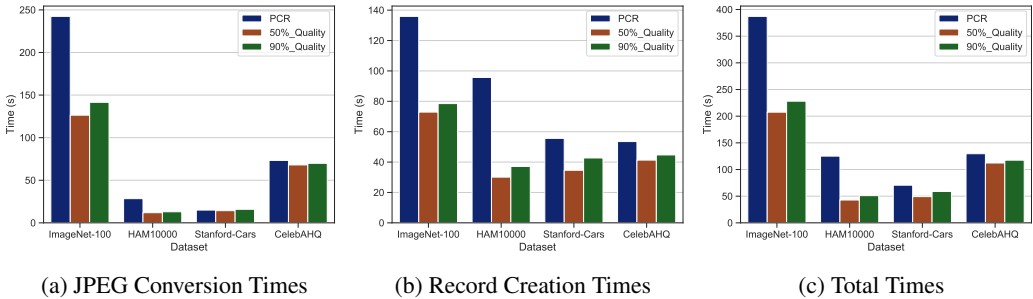

| (a) JPEG Conversion Times | (b) Record Creation Times | (c) Total Times |

Figure 14: Encoding times for baseline JPEG re-encoding and the PCR approach. *Total Time* is the combination of *JPEG Conversion Time* and the *Record Creation Time*. *JPEG Conversion Time* is the amount of time required to convert the JPEG to progressive form or re-encode it to a lower quality JPEG. *Record Creation Time* is the amount of time required to write the images to record format.

different dataset qualities over the course of training (e.g., training with a course-to-fine quality schedule), and, thus, a single training session may consume dozens of distinct dataset qualities.

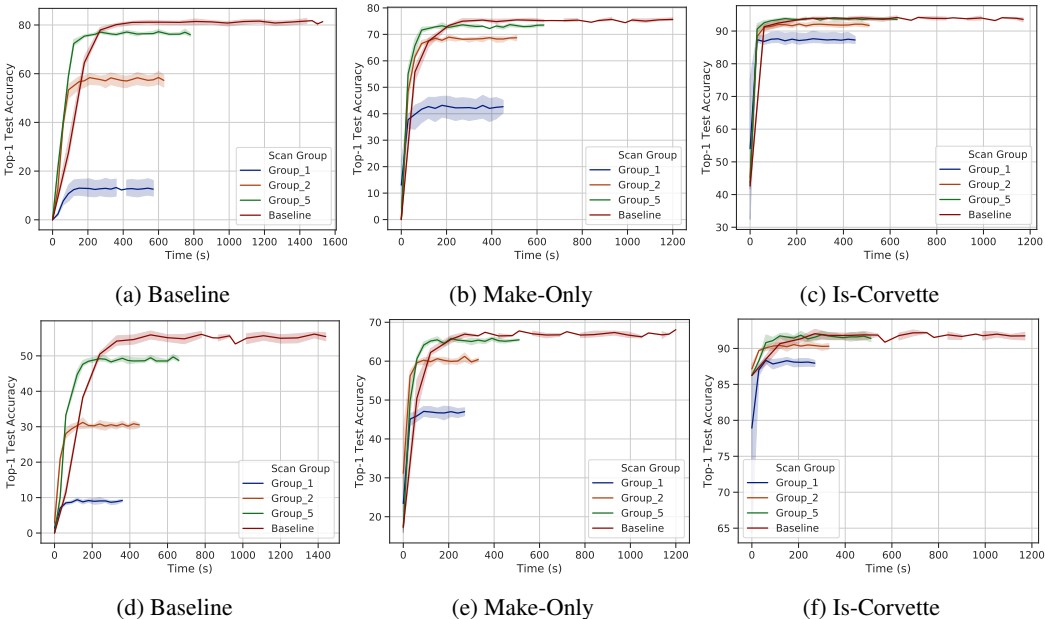

| (a) Baseline | (b) Make-Only | (c) Is-Corvette |
| (d) Baseline | (e) Make-Only | (f) Is-Corvette |

Figure 15: Training accuracy with ResNet-18 (**top**) and ShuffleNetv2 (**bottom**) on a coarser version of the Stanford Cars dataset. The full range of classes is used for *Baseline* (i.e, car make, model, and year create a unique class), only car make is used for *Make-Only*, and a binary classification task of Corvette detection is used for *Is-Corvette*. The gap between scan groups closes as the task is made more simple. Time is the x-axis (seconds) and is relative to first epoch. 95% confidence intervals are shown.

## A.7 IMAGENET-1000 RESULTS

We provide the full ImageNet (i.e., 1000 classes) results with ResNet-18 and ShuffeNetv2 in Figure 17. Only group 5 and the baseline are shown, since lower group numbers have difficulty achieving baseline accuracy parity. The results show that PCRs can speed up training by a factor of 2 while retaining accuracy even with large scale (i.e., over 1 million samples) training.

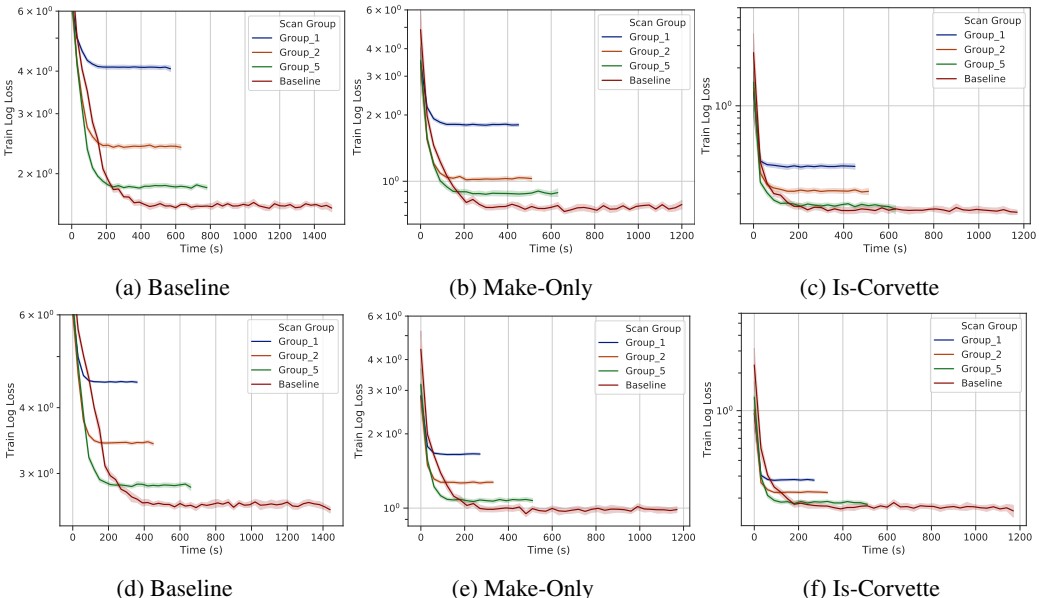

(a) Baseline         (b) Make-Only         (c) Is-Corvette

(d) Baseline         (e) Make-Only         (f) Is-Corvette

Figure 16: Training loss with ResNet-18 (**top**) and ShuffleNetv2 (**bottom**) on a coarser version of the Stanford Cars dataset. The full range of classes is used for *Baseline* (i.e, car make, model, and year create a unique class), only car make is used for *Make-Only*, and a binary classification task of Corvette detection is used for *Is-Corvette*. The gap between scan groups closes as the task is made more simple. Time is the x-axis (seconds) and is relative to first epoch. 95% confidence intervals are shown.

## A.8 IMAGE EXAMPLES BY SCAN

We provide image examples from each dataset that illustrate each scan group in Figure 18. Reading more scans, and, thus, data, from a progressive image results in higher fidelity images, but there are diminishing returns. Images can use a remarkably low amount of scan groups without impacting visual quality, which manifests in bandwidth savings if used accordingly.

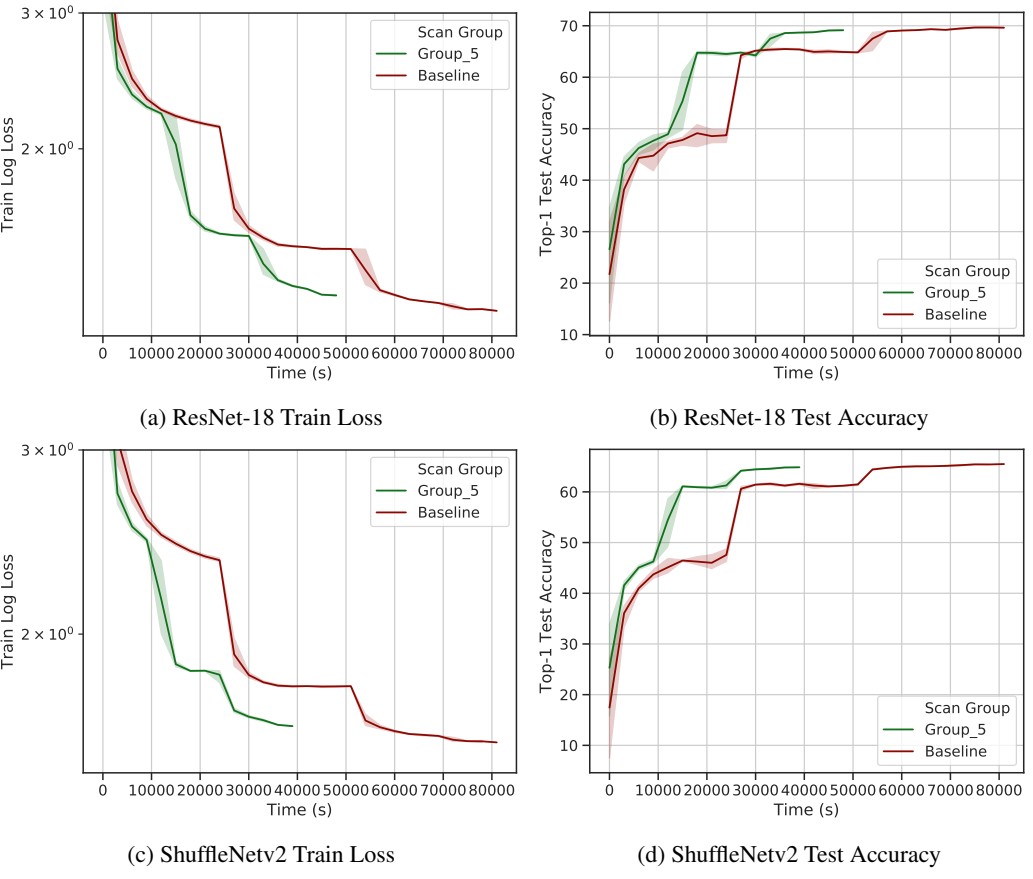

(a) ResNet-18 Train Loss

(b) ResNet-18 Test Accuracy

(c) ShuffleNetv2 Train Loss

(d) ShuffleNetv2 Test Accuracy

Figure 17: Training loss and test accuracy with ResNet-18 (top) and ShuffleNetv2 (bottom) on the 1000 class ImageNet Dataset. Time is the x-axis (seconds) and is relative to first epoch. 95% confidence intervals are shown.

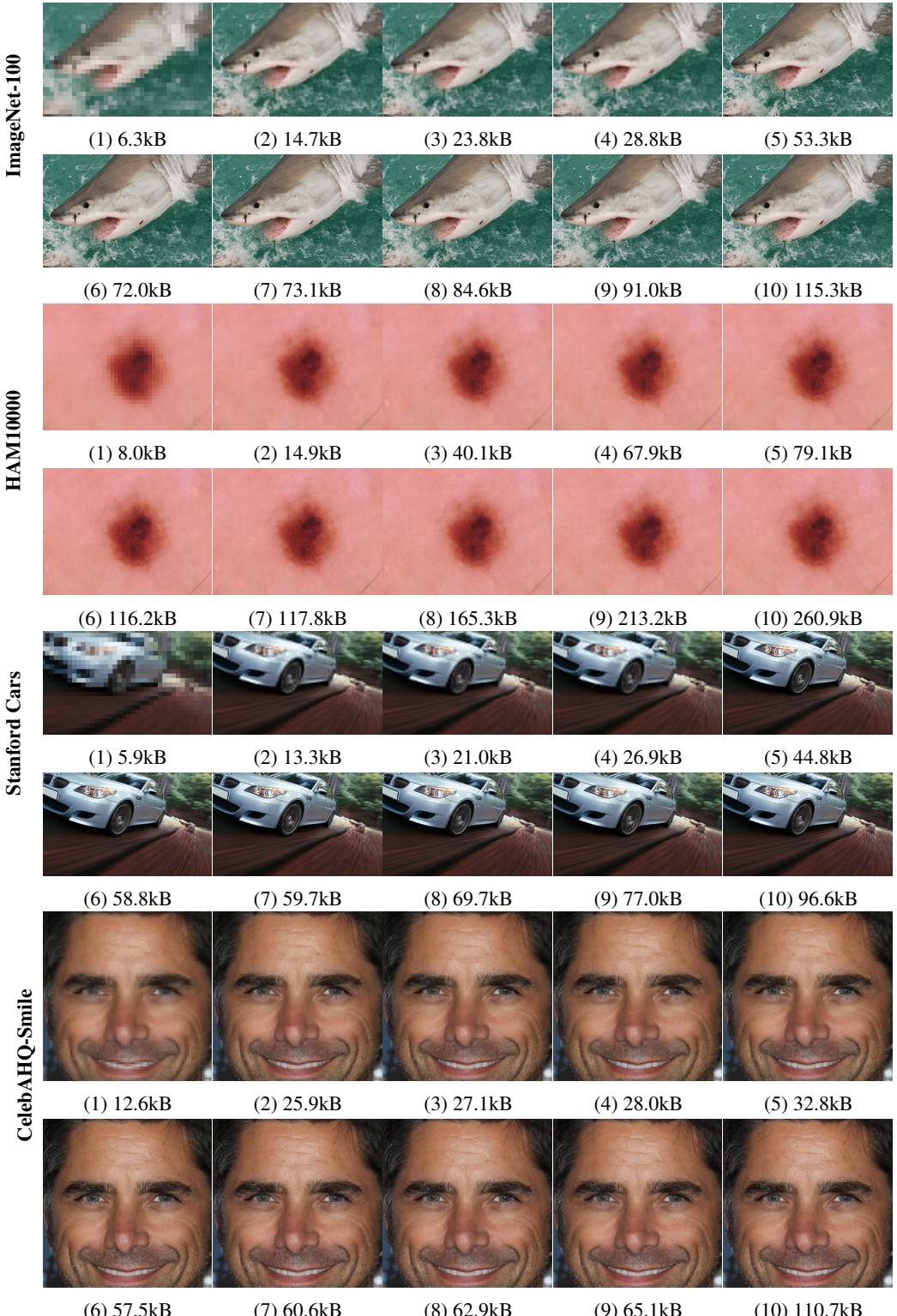

Figure 18: Examples of scans with the corresponding file size. Images are center cropped for demonstration. The amount of scans needed to hit an acceptable level of fidelity is small. Having a larger final size (e.g., lossy compression is minimized) results in more savings for earlier scans.

