# OpenReview forum: "Progressive Compressed Records: Taking a Byte Out of Deep Learning Data"
_ICLR.cc/2020/Conference — Reject_

### Official Review · AnonReviewer3 · 2019-10-24
**Official Blind Review #3**

**Rating:** 6

**Review:**

Summary: This paper introduces a new storage format for image datasets for machine learning training. The core idea is to use progressive JPEG to create sequential scans of the input image, from lower resolution to higher resolution. The authors found that on some datasets, using half of the scans is already enough to reach similar accuracy but speeded up the convergence by a factor of 2.

Detailed feedbacks:
-	The paper presents a simple idea that directly uses the nature of JPEG compression. The paper shows that it can work well and can be potentially integrated into real machine learning dataset storage applications.
-	Related work section is thorough.
-	The experiments are limited to image classifications, and some of the datasets are subsampled (e.g. ImageNet and CelebA). This may not well represent real machine learning tasks, and practitioners may be unsure about the reliability of the compression. The “Cars” dataset contains fine-grained classification, in which the proposed method is
-	Figure 1 is not very clear what is the key advantage of the proposed method, and what are the different mechanisms.
-	Alternatively, one can subsample the pixels and store incremental subsets of those pixels. It would be good if the paper can discuss about this baseline.
-	The data storage format is only loosely related to the main goal of the paper, which is to show that network can still train very well and even faster when receiving partial input data. Once they figured out the number of scans needed for this application, they don’t necessarily need to keep a full lossless version and can just go for a lossy version. In other words, the experiment section can be replaced by any other lossy compression by varying the compression ratio.
-	In my opinion, there could be two reasons for faster convergence. 1) lowered image quality makes the data easier to learn and 2) the smaller data size allows faster reading of data from disk. The paper only shows wall-clock speed-up, but it is unclear which factor is bigger. 2) can be potentially addressed by faster disk reading such as SSD or in-memory datasets. One of the motivations is to help parallel training of dataset and it is also mentioned how non-random sampling of data can hurt training performance. It would be good to showcase how the proposed method can help in those parallel training settings.

Conclusion: This paper presents a simple and effective idea and can be potentially beneficial. However, my main concern is whether the experiments can be representative enough for large scale experiments (e.g. using non-subsampled ImageNet dataset with parallel training using SSD storage). Therefore, my overall rating is weak accept.

**Experience Assessment:**

I have read many papers in this area.

**Review Assessment: Checking Correctness Of Derivations And Theory:**

N/A

**Review Assessment: Checking Correctness Of Experiments:**

I assessed the sensibility of the experiments.

**Review Assessment: Thoroughness In Paper Reading:**

I read the paper at least twice and used my best judgement in assessing the paper.

---

> ### Author Response · Authors · 2019-11-14
> **Revisions To Paper Uploaded**
>
> Thank you for your detailed review and feedback. We respond to each point in more detail below.
>
> [Dataset Limitations]
> We understand the concerns with generalization of our method to other datasets. To address this, we have run our experiments on the full ImageNet dataset and have provided those results in Appendix A.7 “ImageNet-1000 Results”. We want to clarify that our work uses the full CelebAHQ dataset (without subsampling). We have also added coarse-grained vs. fine-grained classification results for the Cars dataset in Appendix A.6 “Coarse Grained vs. Fine Grained Cars Experiments” to address some of the concerns of compression on convergence. Finally, we’ve added dataset details in Appendix A.4 “Dataset Details” to highlight differences in compression quality across datasets. Stanford Cars, for instance, is already highly compressed, and, thus, has reduced benefits from lower scan groups.
>
> In general, our experiments point toward a dependence on the difficulty of the task, regardless of the dataset. Our results are stable across models, and combined with diagnostics like MSSIM, we believe practitioners will be able to properly gauge the appropriate scan group without much tuning. As we discuss in more detail below, the benefit of our approach is that when multiple levels of compression may be appropriate, PCRs obviate the need for storing multiple compressed copies of the dataset.
>
> [Alternatives to JPEG]
> Any progressive image format can work with our method, such as neural network compression. This means, as you noted, that we can use progressive variants of lossless codecs, such as PNG, by subsampling pixels (an interlaced format). However, we chose not to as PNGs have larger file sizes than JPEG (roughly 10x larger, which would conflict with any subsampling gains).
>
> [Static Compression]
> While it is true that part of our contribution is measuring the effect of compression on training, there are cases where dynamic compression is important. We’ve provided conversion times in Appendix A.5 “Record Format Conversion Times” to highlight how costly conversions can be. With many tasks (and thus many distinct compression settings), these conversions can be costly in terms of time. Additionally, practitioners must store at least 2 copies of the dataset (and likely more): one full quality variant (to compress datasets from) and one “accelerated” variant that is compressed. PCRs can provide the same benefits without having multiple copies of the dataset. Further, some training tasks may benefit from varying compression needs at runtime (e.g., Progressive GAN training, which uses multiple distinct resolutions of the same dataset https://github.com/tkarras/progressive_growing_of_gans#preparing-datasets-for-training ); these tasks will necessarily require multiple copies per training session.
>
> [Convergence Diagnosis]
> The reviewer makes a good point that compression can potentially act as a regularizer and improve generalization. However, for our experiments, we do not find this to be the case: When measuring the convergence in terms of accuracy per epoch (rather than time), we observe that lower quality images reduce convergence speed. In contrast, we do observe speedups when measuring wall-clock time, suggesting that the improvements are coming from the reduction in access time and not from improved generalization. We include these accuracy vs. epoch results in Appendix A.1 “Loss, Space Savings, And Accuracy per Epoch”.
>
> Our method attempts to lower training expenses (specifically related to storage bandwidth), which we believe will allow practitioners to utilize the benefits of large-scale machine learning. While alternative techniques (e.g., dataset partitioning) do reduce the randomness in sampling (as discussed in a different reviewer comment (#4)), we view these methods as orthogonal to our technique, which saves bandwidth.

---

### Official Review · AnonReviewer2 · 2019-11-06
**Official Blind Review #2**

**Rating:** 3

**Review:**

The paper proposes using progressive encoding of images and re-arrange of data blocks in images to improve reading speed and therefore training speed.

To fully analyze the maximum possible speed of training, it would be great to the measure upper bound of images/sec, when avoiding reading from disk and just using images from memory.

Decoding a typical progressive JPEG image usually takes about 2-3 times as much time as decoding a non-progressive JPEG, for full resolution, analyzing the time to read vs time to decode the images would be great. It is not clear how changing the number of total groups would affect the image size and the reading speed.

Based on the current experiments it is not clear what is the impact of the batch size when creating PCRs and when reading the image blocks, or the impact of the batch size on the training speed.

Figure 3 is really hard to read and compare times to convergence, authors should provide a table with times to X% accuracy.  Although time to convergence is the key metric, it would be great to know the difference in images/sec of different settings.

Using ImageNet 100 classes (not clear how the 100 classes were chosen) instead of the usual 1000 classes, can distort the results, since it is not clear if higher resolution would be needed to distinguish more classes or not.

Have the authors considered other image compression formats like WebP? How tie is the proposed record encoding with the image compression?

**Experience Assessment:**

I have published in this field for several years.

**Review Assessment: Checking Correctness Of Derivations And Theory:**

I assessed the sensibility of the derivations and theory.

**Review Assessment: Checking Correctness Of Experiments:**

I carefully checked the experiments.

**Review Assessment: Thoroughness In Paper Reading:**

I read the paper at least twice and used my best judgement in assessing the paper.

---

> ### Author Response · Authors · 2019-11-14
> **Revisions To Paper Uploaded**
>
> Thank you for your detailed review and feedback. We have updated the paper to address your feedback on understanding the hardware limits of the used GPUs, image decoding, experiment configuration (i.e., batch size and ImageNet subsampling), clarity of figures/data, and alternatives to JPEG. Below, we provide a detailed response for each item individually.
>
> [Hardware Limitations]
> We have added a section in the appendix (A.3 Experiment Setup) that shows the image rates achievable with in-memory training. Our loading rates approach these limits as the number of scans approaches 1, which suggests that the workload becomes bottlenecked by model updates (rather than the data pipeline) as the number of scans is reduced.
>
> [Image Decoding]
> It is correct that decoding progressive formats can be more expensive than non-progressive formats; please note that this overhead is already taken into account in our time-to-accuracy experiments. However, as per the reviewer’s request, we have added a section in the appendix (see Section A.3 “Experiment Setup”) to directly compare baseline JPEG decoding vs. progressive JPEG decoding rates. We observe less than 50% increase in decoding time using 10 scans; future work can look into how to reduce this overhead further. We keep the number of groups constant (the default 10 scans), but progressive compression does not drastically impact file size as the information content is merely re-arranged. A simple way to reduce overhead in the existing implementation is to reduce the number of scans from 10 to 4, as that is all we use for experiments.
>
> [Batch Size]
> The PCR format is not dependent on the batch size used for training because the same number of images is used for every batch. The number of bytes per batch, however, is reduced. This opens up opportunities, such as increasing the batch size while keeping the number of bytes per batch constant. In our experiments, we typically read multiple (e.g., 10) mini-batches from one record. Longer records are desirable because storage delivers longer, sequential reads faster.
>
> [Time to Accuracy and Image Rates]
> Thanks for these suggestions to further clarify our experiments. We have included a table indicating time to convergence (see Section A.2 “Time to Convergence Table”) as well as images per second of various scan groups (see Section A.3 “Experiment Setup”).
>
> [ImageNet]
> We have added a section in the Appendix (see Section A.4 “Dataset Details”) explaining how the 100 class ImageNet was subsampled (it was done in alphabetical order). As noted above, we have also run our experiments on the full ImageNet dataset and provided those results in Appendix A.7 ImageNet-1000 Results. The 1000 class results mirror the 100 class results; we see a 2x speedup even with a larger dataset.
>
> [Alternatives to JPEG]
> We thank the reviewer for helping to point out the generality of our approach. Indeed, any progressive image format, such as neural network compression, can work with our method. While Webp is not currently progressive, PCRs could utilize WebP if that capability was added. The main issue with other formats is existing infrastructure to support them; JPEG is widely used and optimized, and thus developer efforts required to use it in this setting are reduced. As noted by Reviewer #3, we can use progressive variants of lossless codecs, such as PNG, by subsampling pixels (an interlaced format), but we chose not to as PNGs have larger file sizes than JPEG.

---

### Official Review · AnonReviewer5 · 2019-11-07
**Official Blind Review #5**

**Rating:** 6

**Review:**

This paper introduces Progressive Compressed Records (PCR) which is an on-disk format for fetching and transporting training data in an attempt to reduce the overhead storage bandwidth for training large scale deep neural networks. This is a well written paper that includes all the required background and related works, as well as an easy-to-understand example that runs through the manuscript, explaining what the reader needs to know in order to appreciate the work. The empirical results of several experiments show that the PCR requires up to two times less storage bandwidth while retaining model accuracy.

My only concern is that although the related work section provides a thorough survey of the current methods in the literature, the authors did not demonstrate the performance of state-of-the-art and compare their performance with them. I believe this is necessary to truly validate the superiority of their method over state-of-the-art.


**Experience Assessment:**

I do not know much about this area.

**Review Assessment: Checking Correctness Of Derivations And Theory:**

N/A

**Review Assessment: Checking Correctness Of Experiments:**

I assessed the sensibility of the experiments.

**Review Assessment: Thoroughness In Paper Reading:**

I read the paper at least twice and used my best judgement in assessing the paper.

---

> ### Author Response · Authors · 2019-11-14
> **Revisions To Paper Uploaded**
>
> We thank the reviewer for their positive assessment of our work.
>
> In terms of state-of-the-art, we are not aware of any other works that use progressive compression in the context of machine learning, and to the best of our knowledge, the intersection of storage and machine learning is also poorly explored.
>
> Current methods typically avoid disk I/O entirely by caching datasets in RAM. While this is a viable approach for small datasets, not all datasets can fit in RAM and thus we focus on large datasets. To the best of our knowledge, TFRecord and RecordIO can be considered state-of-the-art in the large dataset domain. Both TFRecord and RecordIO are implementations of the same idea, which is a record layout. For this reason, we compared against full-quality images stored in batch form (i.e., record layouts), which resembles state-of-the-art record formats (e.g., RecordIO, TFRecord) without relying on the implementation details of those highly-engineered formats, which may obscure the comparison.

---

### Official Review · AnonReviewer4 · 2019-11-07
**Official Blind Review #4**

**Rating:** 3

**Review:**

The paper demonstrates an interesting application of progressive compression to reduce the disk I/O overhead of training deep neural networks. The format encodes the trade-off between data fidelity and I/O bandwidth demand naturally, which could be useful when I/O is the bottleneck.

My major concern is that the paper should be clearer about the setting.
* Does your work target the case where data cannot fit in RAM and should be fetched from local disk or through network? However, the datasets used in the evaluation look small and could fit in RAM.
* How are mini-batches created? You mentioned in the related work that previous work (Kurth et al., 2018) lets each worker sample from a local subset instead of performing a true sampling of the whole dataset. Does your work perform true sampling? How much benefit does it give?
* Is disk I/O really a bottleneck in training? There are many evidence [1][2][3] of almost linear scalability in training ResNet on *full* imagenet across hundreds or even thousands of GPUs. These work focus heavily on network communication rather than disk I/O. Does your setting differ from theirs? How does your approach compare with their techniques for optimizing disk I/O?

That being said, I think this approach should be appealing when the I/O bandwidth is limited and dynamic. Examples include training on edge devices, or federated training where data needs be fetched via ad-hoc network.

Other detailed comments:

* Figure 1 is not very informative and quite puzzling. There is no definition of quality at that point.
* Sec 2 paragraph 3. What is the issue of data augmentation with the standard JPEG compression? Does your compression ease data augmentation?
* Sec 3.1 paragraph 1. "This is turn enables ..." -> "This in turn enables ..."
* How to decide the number of scans? Does it have impact on the I/O efficiency?
* Evaluation
  - I'm not familiar with Ceph. Why choose this particular environment? Does it bring in extra overhead (e.g., communicating with metadata server). What does the network topology look like? Is the data loading stall (figure 7) due to network congestion?
 - It worth evaluating more tasks such as detection and segmentation to measure the impact of compression.


[1] Accurate, Large Minibatch SGD: Training ImageNet in 1 Hour, Goyal et al.
[2] Massively Distributed SGD: ImageNet/ResNet-50 Training in a Flash, Mikami et al.
[3] Image Classification at Supercomputer Scale, Ying et al.

**Experience Assessment:**

I have published one or two papers in this area.

**Review Assessment: Checking Correctness Of Derivations And Theory:**

I assessed the sensibility of the derivations and theory.

**Review Assessment: Checking Correctness Of Experiments:**

I carefully checked the experiments.

**Review Assessment: Thoroughness In Paper Reading:**

I read the paper thoroughly.

---

> ### Author Response · Authors · 2019-11-14
> **Revisions To Paper Uploaded**
>
> Thank you for your detailed review and feedback. We have uploaded changes addressing the issues you raise, including updating Figure 1 to provide a more intuitive description of the setting of this work. Other changes are described below.
>
> [Small Datasets]
> Yes, our work targets large datasets that cannot fit in RAM. We used smaller datasets in order to rigorously evaluate our method within our computational budget. However, we understand the importance of validating the method at scale, and have therefore also run our experiments on the full ImageNet dataset and provided those results in Appendix A.7 “ImageNet-1000 Results”. Thank you for this suggestion.
>
> [Sampling]
> Previous work uses data partitioning to allow each worker to hold a subset of the dataset. Data partitioning is an orthogonal optimization to ours; one can use PCRs with data partitioning. However, some users of data partitioning (e.g., Kurth et al., 2018) apply an additional optimization on top, which is to cache the partitions in memory or on a local SSD. In particular, they partition the 3.5TB dataset such that an 800 GB SSD can store each partition; this means each worker samples from a subset of the dataset (i.e., there is 0 probability of sampling certain images for any fixed worker). While sacrificing some sampling guarantees is common (e.g., record formats do it by correlating the samples drawn within records), static partitioning is one of the more extreme tradeoffs and relies on creating representative partitions. Additionally, static partitioning only works when the data is small enough to fit in a cluster’s aggregate memory or fast storage. In this case, 800 GB per node represents a hard limit after which a distributed file system would have to be used.
>
> [ImageNet Training]
> It is common practice to shard the dataset among each node, which allows the dataset to be collectively stored in a fast cache. For instance, Mikami et. al. 2018 state that data is partitioned between each worker (see Remark 4). Ying et al. 2018 do not report exactly how they do the data movement, but they have 409 TB of RAM among the utilized nodes, while ImageNet is only 150GB. If data is really in memory (or on a local fast SSD), then there are less concerns for I/O bottlenecks. However, as noted above, this approach is limited by the available space for storing the dataset.
>
> [Data Augmentation]
> We mentioned data augmentation in the paper to note that some data augmentation methods degrade image quality (e.g., with random noise or blur). If the image quality is already degraded, then it is intuitive that more compression may be tolerable during training (e.g., compression followed by blur would look similar to non-compressed followed by blur). Similarly, downscaling an image can reduce the artifacts introduced by compression (e.g., a low quality 4K image can be resized to a high-quality 256x256 image). Therefore, we are simply noting that some tasks may tolerate higher levels of compression given a particular set of data augmentations.
>
> [Number of Scans]
> We use 10 scans as that works for our experiments and is the default number used by the transcoder that we use. Adding more scans allows one to more finely trade off image quality vs. I/O bandwidth, but in practice we do not expect needing more than 10 scans (in fact, we only use 4 distinct settings).
>
> [Ceph]
> We use Ceph as it is an open source, widely deployed distributed store. Ceph’s metadata overhead is limited, as each node can determine what node the data is stored on. It’s also worth noting that by using record formats, we are only accessing a relatively small number of distinct files (rather than many individual images). Thus, the metadata overheads are less of a concern. We use a 40Gb Ethernet network to connect all nodes, which should be sufficient for training at this scale.
>
> [Additional Tasks and Modalities]
> We agree that additional tasks (e.g., segmentation) and modalities (e.g., audio, video, text) would be interesting directions for future work. For this work, we decided to focus on deepening our understanding of object classification by identifying dataset properties which favor compression, such as easy vs. hard tasks (Appendix A.4 “Dataset Details”, A.6 “Coarse Grained vs. Fine Grained Cars Experiments”). Some of this understanding would hopefully transfer to the other tasks.

---

### Author Response · Authors · 2019-11-14
**Revisions To Paper Uploaded (Summary)**

We thank the reviewers for their time and helpful feedback. After carefully considering reviewers’ comments, we made revisions to our paper to address concerns, and have uploaded an updated version of the paper. A summary of major changes follows:

* We have run our experiments using the full 1000 class ImageNet dataset (see Section A.7 “ImageNet-1000 Results” in our updated submission). Our results indicate that the PCR approach generalizes to larger datasets.

* We’ve added further motivation (see Section A.5 “Record Format Conversion Times”) for this work, including encoding times as well as an updated Figure 1. PCRs offer a natural trade-off between speed and quality, and do so without sacrificing space or time per task.

* We’ve added microbenchmarks (see Section A.3 “Experiment Setup”) pertaining to various stages of the pipeline (e.g., computer speeds, decoding overheads). We include the rates of both training (in terms of images/second) and decoding to highlight the potential bottlenecks of the system. The results show that while image decoding does add overhead (up to 50% in terms of decoding time), the slowdown does not prevent achieving a training rate close to the maximum possible.

* We’ve added details pertaining to the datasets used in Section A.4 “Dataset Details”. Since our approach is dependent on the JPEG quality of the dataset, we additionally include quality estimates for each dataset to highlight each dataset’s compressibility. We’ve further analyzed the Cars dataset to determine conditions when a dataset would be coarse-grained enough for PCRs to be helpful (see section A.6 “Coarse Grained vs. Fine Grained Cars Experiments”). We hope this analysis allows practitioners to reason about the conditions under which image degradation (from compression) is safe. To the best of our knowledge, this is the first work investigating how to calibrate compression parameters for training.

* We’ve added accuracy vs. epoch plots to disentangle the statistical effects of PCRs from the system speedups (see section A.1 “Loss, Space Savings, And Accuracy per Epoch”). Our results indicate that most of the improvements come from faster image rates.

Once again, thank you for your valuable feedback!

---

### Decision · Program_Chairs · 2019-12-19

**Decision:**

Reject

**Comment:**

Main content: Introduces Progressive Compressed Records (PCR), a new storage format for image datasets for machine learning training.
Discussion:
reviewer 4: Interesting application of progressive compression to reduce the disk I/O overhead. Main concern is paper could be clearer about setting.
reviewer 5: (not knowledgable about area): well-written paper. concern is that related work could be better, including state of the art on the topic.
reviewer 2: likes the topic but discusses many areas for improvement (stronger exeriments, better metrics reported, etc.). this is probably the most experienced reviewer marking reject.
reviewer 3: paper is well written. Main issue is that exeriments are limited to image classification tasks, and it snot clear how the method works on larger scale.
Recommendation: interesting idea but experiments could be stronger. I lean to Reject.